# Membrane Dialysis for Partial Dealcoholization of White Wines

**DOI:** 10.3390/membranes12050468

**Published:** 2022-04-26

**Authors:** José Ignacio Calvo, Jaime Asensio, Daniel Sainz, Rubén Zapatero, Daniel Carracedo, Encarnación Fernández-Fernández, Pedro Prádanos, Laura Palacio, Antonio Hernández

**Affiliations:** 1Institute of Sustainable Processes (ISP), Dr. Mergelina s/n, 47071 Valladolid, Spain; 2Departamento de Física Aplicada, ETSIIAA, Universidad de Valladolid, Avenida de Madrid 57, 34004 Palencia, Spain; ppradanos@uva.es (P.P.); laura.palacio@uva.es (L.P.); antonio.hernandez@uva.es (A.H.); 3Área de Tecnología de los Alimentos, Escuela Técnica Superior de Ingenierias Agrarias, Universidad de Valladolid, Avenida de Madrid 50, 34004 Palencia, Spain; jaimeasensioo@gmail.com (J.A.); dsainze@gmail.com (D.S.); rd.dr91@gmail.com (R.Z.); carracedoesguevillas@gmail.com (D.C.); encarnacion.fernandez@uva.es (E.F.-F.); 4Departamento de Física Aplicada, Facultad de Ciencias, Universidad de Valladolid, Paseo de Belén 7, 47011 Valladolid, Spain

**Keywords:** wine dealcoholization, nanofiltration, pervaporation, dialysis, white wine

## Abstract

Membrane dialysis is studied as a promising technique for partial dealcoholization of white wines. The performance of three membrane processes applied for the partial dealcoholization of white wines of the Verdejo variety has been studied in the present work. Combination of Nanofiltration with Pervaporation, single step Pervaporation and, finally, Dialysis, have been applied to white wines from same variety and different vintages. The resulting wines have been chemically and sensorially analyzed and results have been compared with initial characteristics of the wines. From the results obtained, we can conclude that all procedures lead to significant alcohol content reduction (2%, 0.9% and 1.23% *v*/*v* respectively). Nevertheless, the best procedure consists in the application of Dialysis to the wines which resulted in a reasonable alcohol content reduction while maintaining organoleptic properties (only 14 consumers were able to distinguish the filtered and original wines, with 17 consumers needed to be this differences significant) and consumer acceptability of the original wine. Therefore, membrane dialysis, as a method of partial dealcoholization of white wines, has undoubted advantages over other techniques based on membranes, which must be confirmed in subsequent studies under more industrial conditions. This work represents the first application of Dialysis for the reduction of alcohol content in wines.

## 1. Introduction

In recent years, most world winemaking regions have found that climatic change, leading to generalized global warming and important reduction in total rainfall, resulted in severe problems for the production of quality wines, among which: difficulties to achieve a correct balance between phenolic and technological grape maturation; grapes raisin before reaching a good maturation; accumulation of aromas in the white grape berry, more frequently in cold climates [1] and finally, resulting musts which have lower acidity, thus an excessive pH, and higher sugar contents with variation in the concentration of polyphenols [2].

This increase of sugar in must lead to wines with higher alcoholic content [3,4], which also increases the perception of a health risk from consumers, [5], a sensitivity that also could be supported by the restrictive legislation on alcohol rates allowed when driving. An excessive alcohol content also affects the taxes associated with the exportation of wines, making them difficult to compete in markets that are already very saturated and to consumers who traditionally have little appetite for high alcohol content beverages.

Of course, premature grape harvest and winemaking could reduce the potential alcohol content of the berries. Nevertheless, this would affect the final wine quality, since the phenolic maturity would not be yet fully achieved leading to more acid and less colored wines, at least in the case of red wines [6].

To reduce the alcoholic content of wines, different techniques can be used. The points to be met in order to carry out partial dealcoholization are, [7]:

The decrease in the acquired alcoholic strength by volume cannot be greater than 20% vol of original wine and final product must meet the definition of wine.

The treated wines must be suitable for direct human consumption (not having organoleptic defects).

If operations have been carried out to increase the alcoholic degree in one of the winemaking input products, the alcohol could not be eliminated.

Studies developed in the field of viticulture show strategies to act on the techniques of wine cultivation (irrigation or plant canopy management to control photosynthesis), thus avoiding the excessive accumulation of sugar [8]. There are even techniques that are performed before alcoholic fermentation, such as the addition of glucose-oxidase and catalase enzymes [9].

Many studies have also evaluated the suitability of non-*Saccharomyces* and *Saccharomyces cerevisiae* yeast strains that could prevent ethanol formation in alcoholic fermentation [10,11,12,13,14,15].

Several publications have successfully studied a pre-fermentation approach by membrane Nanofiltration (NF) for sugar control in grape must [6,16,17,18,19]. Results showed that the best NF technique was a two-stage NF process [18], which promoted higher recovery of grape must chemical properties and fewer volume losses. However, sensory evaluation of the wines obtained showed that they were less persistent in mouth and had lower flavor in comparison to control wines. This aroma depletion was attributed to the possible loss of volatile compounds (primary aroma compounds) during grape must nanofiltration.

In order to minimize those losses, primary aroma compounds could be recovered, before NF, from the grape juice and added to the filtered must before fermentation. Traditional aroma recovery processes such as distillation, adsorption and solvent extraction are not useful since the operation at high temperature [20] could deteriorate original juice and its aromas. In view of their intrinsic characteristics, namely high selectivity and possibility of operation at moderate temperatures, Pervaporation (PV) is a membrane process that seems highly appropriate for the separation of dilute species in liquid solutions [21,22,23]. Specifically, organophilic PV membranes have a high potential for recovering natural and natural-like aroma compounds, highly diluted in complex aqueous media, [24].

On the other hand, non-alcoholic (or reduced alcoholic beverages) can be produced by eliminating ethanol from a fully fermented beverage (post-fermentation approach). The most common separation techniques for the dealcoholization of beverages are thermal treatment or membrane-based processes [25]. The heat treatment processes include evaporation and distillation or steam separation, in both cases under vacuum conditions.

Other techniques as osmotic distillation or rotating cone columns have been successfully used [26,27,28,29,30].

Membrane-based processes include Electrodialysis (ED) mainly used for tartaric stabilization, acidification, and de-acidification of musts and wines, [31]. While the membrane processes used for alcohol reduction of beverages include Reverse Osmosis (RO) [32,33], NF [34], and PV [35].

Among membrane separation processes, Dialysis (D) offers interesting features which suggest that its application for the partial reduction of alcohol in wines could be highly beneficial. For example, it does not use any pressure or temperature gradient, which could result in sensorial worsening of the resulting product. It has been successfully applied for alcohol reduction of beers [36]. Nevertheless, to the best of our knowledge, no study has been done to apply Dialysis to wine dealcoholization [37].

Therefore, the scope of the present research is to compare the feasibility of several approaches for the reduction of alcohol content in a white wine. Three distinct approaches (from more conventional Nanofiltration to less aggressive Pervaporation or Dialysis) have been tested on similar white wines and features and differences of the resulting wines will be analyzed in terms of alcohol reduction, chemical properties, and organoleptic descriptors.

## 2. Materials and Methods

### 2.1. Dealcoholization Procedures

Three procedures of membrane-based wine dealcoholization will be compared in this work: (a) The first approach consisted, as previously commented, in using a NF membrane to reduce the alcohol content of the treated wine. NF permeate resulted in a significant reduction of alcohol content but also an important decrease of aromatic compounds. Therefore, permeate was further pervaporated to recover some of the aromas lost during NF. From the retentate of NF and the aromas recovered by PV, the starting wine is reconstructed, from which a sensible reduction of the alcohol content is expected, as well as an aromatic and organoleptic composition, as close as possible to the original wine. (b) A second approach consisted in a one-step filtration. Wine was subjected now to a PV step, from which part of the alcohol of the wine was recovered as permeate. Also, some aromas of the wine were pervaporated and passed through the membrane. Since most aroma compounds are more volatile than alcohol, the initial fraction of PV permeate was rich on aromas and less in alcohol. Consequently, this first fraction was recovered and added to the original wine once filtration finished. The rest of the PV permeate, mostly composed of alcohol, was discarded to get the maximum alcohol degree reduction. (c) The third approach consisted also in a single step filtration, by means of a D setup. Wine and pure distilled water were recirculated from both sides of an appropriated membrane by means of two peristaltic pumps in no transmembrane pressure conditions. Two transport mechanisms acting simultaneously (osmotic flow of water to the wine side and diffusion of volatile compounds of wine, mainly alcohol, to the water side) lead to a resulting wine with lower alcohol content.

Consequently, three experimental devices were designed to perform the NF, PV, and D stages, respectively—devices that will be described below.

### 2.2. Filtration Devices

#### 2.2.1. Nanofiltration Experimental Set Up

NF was carried out in an automated pilot plant (see Figure 1), designed at our Research Lab (SMAP), and consisting of a 30 L stainless steel feeding tank with a cold jacket (1) for temperature control through a Neslab (ThermoFischer, Madrid, Spain) refrigerator bath (2). Feed fluid was recirculated with a Hydra-Cell D10 pump (Wanner Eng., Minneapolis, MN, USA) (3) to a flat membrane module (4) with transverse flow, housing the membrane (5), with an active membrane area A = 6.6 × 10^−3^ m^2^. Two digital pressure gauges located before and after the membrane module measured the inlet (P_i_) and outlet (P_o_) pressure. In order to manually adjust the pressure inside the module, a needle valve was placed at the output of the unit. The transverse flow was measured with a Tecfluid (Barcelona, Spain) flow meter (measuring range 0–10 L/min) and automatically adjusted to avoid the pressure exceeding a critical level (settled at 40 bar), so that no breakage occurred in the membrane. Tangential speed, inlet and outlet pressure, and permeability measurements are completely automatized.

Feed stream temperature was kept constant at 6 °C during filtration, while pressure was kept controlled at 30 ± 1 Bar and the feed rate was adjusted to 6.0 ± 0.2 L/min. NF permeate was collected in a graduated 2 L beaker to estimate the volume yet permeated.

After finishing the NF step, permeate was kept in the fridge up to complete the PV step.

#### 2.2.2. Pervaporation Experimental Set Up

The recovery of the aroma compounds of NF permeate (Case 1) or the aroma recovery and alcohol elimination (Case 2) was carried out in a laboratory cell equipped with a PV flat sheet. The installation built for the experiments is schematized in Figure 2.

The pervaporation device contains a pervaporation membrane, placed in a flat sheet cell (4) that provides an active area of 6.6 × 10^−3^ m^2^. A Flojet (Xylem, Hertfordshire, UK) electric pump (3), model R2100-232, extracts the wine to be pervaporated from the feed tank (2) of 3 litre capacity and equipped with a cold jacket (1) controlling temperature by a Julabo model F12 (Iberfluid, Madrid, Spain) circulation bath. This fluid circulates tangentially over the membrane to limit the effects of concentration polarization. Two nanoscale pressure gauges (6) are placed before and after the cell containing the membrane to avoid overpressures in the membrane; which can be controlled by a needle valve (5), placed after the outer meter. In order to control the flow rate of the fluid through the circuit, a flowmeter (Parker, Cleveland, OH, USA) of the rotameter type is placed. On the other hand, two nitrogen traps (7) are placed at the exit of the pervaporation membrane cell, prepared to collect permeate. Prior to these traps, a three-way valve (8) is placed to direct permeate towards a liquid nitrogen trap or another. Permeate is then condensed in those 2 liquid nitrogen traps alternately. While one of the traps is tempered, the other continues collecting permeate that passes through the membrane. Between the nitrogen traps and the vacuum pump, another three-way valve is used (9), which allows the suction of this pump to be diverted to one or the other trap. This pump (10) is of the vacuum displacement type of the brand Agilent Technologies (Madrid, Spain) model SH 110, which achieves a minimum pressure of 2 mmHg. The liquid nitrogen traps are kept inside a Dewar vessel in which the nitrogen is poured, delaying the evaporation of it. Food fat was also used in order to facilitate the closure of the flask where permeate is collected, and a 20 mL volumetric pipette to extract permeate without touching the fat.

#### 2.2.3. Dialysis Set Up

The equipment used to perform D step is schematized in Figure 3. It consists of an Eheim (Leadfluid, Hebei, China) peristaltic pump, model 1048 (5) that aspires the wine (1.4 L approximately) from the 2 L Erlenmeyer (1) to the cell (3) containing the dialysis membrane (4), with an effective membrane area of 4.4 × 10^−3^ m^2^. Then it returns to the initial Erlenmeyer flask, so that it recirculates continuously at a given flow rate.

On the other hand, another Erlenmeyer flask (2) contains 1.4 L of a complementary liquid (in our case Milli-Q water, from Merck, Mollet del Vallés, Spain) that is aspirated by a peristaltic pump equal to the one described above (5). It passes in a co-current direction through the membrane cell and collects the dialyzed alcohol from the wine. Finally, the flow rate is controlled by a rotameter type Parker flowmeter (6).

Both Erlenmeyer flasks are covered with Parafilm^®^ to prevent oxidation and loss of aromas and alcohol from the synthetic and real wine. In this way, we reduce the error due to losses that may occur during the process.

### 2.3. Membranes

The NF membrane used in the first case was DK (DK1812C-28D, from Lenntech, Delfgauw, The Netherlands), from the D series, an asymmetric membrane made by phase inversion, from General Electric (GE). The DK membrane has a minimum rejection of 98% over 2000 ppm of MgSO_4_ at 25 °C (77 °F) and 110 psi operating pressure.

The PV membrane used in cases 1 and 2 consisted in a PDMS based organophilic membrane made and commercialized by Pervatech (Rijssen, The Netherlands). Previous work in our lab (23) showed that PDMS based membranes give nice performance for aroma recovery through PV of natural grape must.

Regarding D membranes, to be sure of the optimal alcohol recovery of the process, three suitable membranes were studied for the Dialysis system:

Membrane type PLGC 15005: Marketed by Merck (Madrid, Spain) under the commercial name of Ultracel, it is a UF membrane composed of low binding regenerated cellulose. It has a nominal molecular weight value of 10,000 Da and is presented in the form of discs with a diameter of 150 mm.

X25 SPECTRA/POR membranes: In this work 2 different membranes of this type, supplied by Fisher Scientific (Madrid, Spain), were tested. These membranes come in the form of flat sheets composed of regenerated cellulose and they are aimed for use in dialysis experiments. They present good compatibility with dilute strong acids and bases or concentrated weak acids and bases, many alcohols, and organic compounds. The two different types of membranes tested are: X25 SPECTRA/POR 1, 6-8KD, MWCO 240 × 240 mm sheet, with a nominal molecular cut-off weight (MWCO) of between 6000–8000 Da, and presented as flat sheets with a length of 240 mm and a width of 240 mm; and X25 SPECTRA/POR 2, 12–14KD, MWCO 200 × 200 mm sheet, having a MWCO between 12,000–14,000 Da, also in the form of flat sheets of 200 mm in length and 200 mm in width.

Main nominal characteristics of the used membranes are summarized in Table 1.

### 2.4. Chemical Analysis

After each filtration procedure, the analysis of the main oenological parameters were carried out on both wines, the original wine and the dealcoholized wine of all tests, nanofiltration, pervaporation, or dialysis.

Chemical parameters as pH, total and volatile acidity, alcoholic degree, total polyphenol index (TPI), total and free sulphur dioxide, and color were determined according to the principles and methods established by the International Organization of Vine and Wine [38]. All the chemical analysis were carried out in duplicate for each wine, being both results averaged. From the data obtained from the chemical analyses, a statistical treatment was carried out by means of analysis of variance (ANOVA). For this, the Statgraphics Centurion XVIII program (StatPoint Technologies, Inc., Warrenton, VA, USA) has been used.

Finally, the time evolution of the PV and D processes was followed by the aid of a differential refractometer, from Atago (Saitama, Japan). Regularly, small quantities (5 mL) were obtained from the wine side and their refractive index measured in that device. Then, using a calibration curve, refractive index measured data were converted in contents of alcohol.

### 2.5. Synthetic Wine

While, for the case of NF or PV, we had previous experience on the suitability of the membranes selected and the expected duration of the experiments, [24], for the case of D, some previous experiments were needed aimed to select the most appropriated membrane.

Then, for the third case, a synthetic wine was prepared and used to test the best performance membranes. Synthetic wine was composed of alcohol at 96% vol. and mostly distilled water (88.6% vol., for an approximate final graduation of 11.4% vol.).

### 2.6. Verdejo White Grape Wine

The wine used in this study consisted in an experimental Verdejo wine elaborated in the Agricultural Engineering School (University of Valladolid, Palencia, Spain). The variety, with origin in the region of Castilla y León, produces full flavored white wines (Verdejo appellation). Different amounts of such experimental wines (coming from different years and harvests) were used in all case studies. For each case, a different number of wine bottles was used.

To homogenize the different bottles used and prevent membrane clogging since the original wines were bottled unfiltered, a similar procedure was followed in all cases: The number of bottles necessary for each experience was selected. Then, all the wine was homogenized in an appropriated tank to facilitate the sedimentation of the thickest particles. Finally, the resulting homogeneous wine was again bottled in 0.75 L standardized bottles and used for the filtration experiments designed for each case. In all cases enough number of bottles remained stored to be used as a reference sample to carry out the subsequent wine analyses and tastings

Case 1: (NF + PV)

2014 Vintage. The initial alcohol content of the wine used in Case 1 was 11.2% *v*/*v*

Case 2: (PV)

2015 Vintage. For this case, initial alcohol content was 11.5% *v*/*v*.

Case 3: (D)

Finally, the wine used in this case (from 2018 Vintage) presented an initial alcohol content of 9.90% *v*/*v*.

In all three cases, the wine was elaborated just after grape harvesting, while filtration procedure, chemical analysis, and sensorial analysis were accomplished in the following year (2015, 2016, and 2019, respectively).

### 2.7. Sensory Analysis

Sensory evaluation of the wines was conducted in three sessions (one for each case of study), using in all cases triangle tests to demonstrate the existence or not of a noticeable difference between products [39], and check-all-that-apply (CATA) questions [40] to characterize organoleptic properties of treated and untreated wines.

The check-all-that-apply (CATA) questionnaire contained 15 terms related to the sensory characteristics of the Verdejo wines grouped in two visual terms (clean, rusty), eight olfactory terms (fruity, citric, tropical, herbaceous, balsamic, reduced, aniseed, aroma intensity), four terms in the mouth (bitter, acid, mouth volume persistence), and finally consumers were asked to give a general evaluation of the wine by stating whether the wines were acceptable for them.

Consumers were asked to check all the terms that they considered appropriate to describe each wine. The terms were selected based on published data [41] considering the descriptors selected by trained assessors in preliminary studies, along with the typical characteristics of Verdejo wines.

A total of eighty-six people participated in the study. The first session was conducted with twenty-six consumers, while twenty-seven participated in the second testing, and thirty-three tested the wines from third case. In all cases, consumers were students and professors of the Oenology degree in the Agricultural Engineering School with experience in wine tasting and ages between 18 and 50.

In all sessions, the samples were served as 25 mL aliquots in standardized wineglasses [42], which were coded with 3-digit numbers randomly generated. The serving temperature of the samples was 10 ± 1 °C. All these sensory evaluations were carried out at the Sensory Science Laboratory of the School of Agricultural Engineering, at the University of Valladolid, Palencia (Spain), in individual booths designed in accordance with ISO 8589 [43].

For the triangle test, the significance level was determined according to the number of tasters who correctly distinguished between the tested samples, as well as the proportion of the population that the sample can distinguish [38].

Frequency of use of each one of the terms of the CATA question was determined by counting the number of consumers that used that term to describe each sample. Cochran’s Q test [44] was carried out to identify significant differences among samples for each of the sensory terms using IBM SPSS Statistics software, version 24.0.

## 3. Results and Discussion

### 3.1. Filtration

#### 3.1.1. Case 1 (NF + PV)

Case 1 consisted in two filtration steps (NF and PV) separately, and a further reconstruction of the wine, so each step will be now commented.

##### Nanofiltration

For the NF process, departing from an initial effective volume of 8 L of wine, after 10 h of nanofiltration, 1700 mL of permeate were collected in a graduated beaker (total capacity 2.0 L). Said permeate was stored in a refrigerator at the end of the NF stage to assure its best conservation until PV step was carried out. During NF filtration, periodic measurements of permeate flux were done showing that membrane permeability was constant during all the process. So, we can conclude that membrane was not appreciably affected by fouling.

##### Pervaporation

Regarding the pervaporation process, it was started with the initial volume of 1700 mL of permeate resulting from the nanofiltration step and, after 520 min, 91 mL of pervaporate were obtained distributed in 12 extractions as shown in Figure 4.

In Figure 4a it can be observed that the slope of the volume of permeate collected versus time was reasonably constant over time; indicating that the membrane at no time was filled and only vapor transport occurred (something otherwise expected when working with gaseous and liquid phases on both sides of the membrane). In fact, the figure also presents a straight line corresponding to the linear fit of the first points. It can be seen that, in the initial moments, the flow is constant due to the passage of aromas and a greater presence of ethanol in the liquid phases, but afterwards, a slight decrease in flow is observed due to the elimination of the most volatile (and organophilic) compounds due to the reduction of alcohol in the liquid phase.

A refractometry follow-up of the alcoholic content of permeate collected can be seen in Figure 4b. The first extraction, which corresponds to the aromas according to their volume [23], corresponds to a concentration of 23.5%. The next extraction clearly increases this alcohol content, followed by a slow decrease up to a final value of 24.5% (after 540 min of PV filtration). The fact that the first point has a lower value may be related to the time-lag of the membrane until the stationary profile is defined, since the refractometer will not differentiate alcohol from other volatiles. These concentrations of ethanol in the permeate are lower than another study in which a wine was dealcoholized by PV [35]; this may be due to the use of a different type of membrane or a starting ethanol concentration of the pervaporated fluid.

Since the content of alcohol in permeate is always higher than original wine, we can be sure that PV retentate alcohol content will decrease during the process. That is the reason why the alcohol content of the permeate extractions is also decreasing due to the decrease in alcohol of retentate which makes the alcohol extraction process less effective with time. This is in line with the decrease in flow observed in Figure 4, which, as we have said, is due to the decrease in the concentration of alcohol in the feed. The same reduction of components collected in PV was observed in another study [45] as the PV process progressed.

Taking into account the alcoholic content of all resulting fractions (11% *v*/*v* for the NF retentate and 6.9% *v*/*v* for PV retentate) the wine was reconstructed to achieve the maximum alcohol reduction permitted. So, 1.47 L of retained NF and 1.36 L of retained PV were mixed and added the 3.8 mL of aromas collected during initial steps of PV. Finally, 2.83 L of dealcoholized wine were obtained with a final alcohol content of 9.2% *v*/*v*.

#### 3.1.2. Case 2 (PV)

Next, filtration results coming from the second approach will be discussed.

##### Pervaporation

For this process, a total volume of 2.5 L of the control wine was filtered. The PV process lasted for 526 min (8.8 h) and after that a volume of 21.8 mL of PV permeate was obtained, distributed in 15 extractions, whose cumulative volumes can be seen in the following graph (Figure 5a).

Similarly to that arisen in the PV step of Case 1, this figure allows for the assurance that the membrane was never liquid filled. The hydrophobicity of the PDMS membrane, succeeded in preventing the passage of aqueous solutions. So, only the volatile substances present in the medium, favored by the vacuum created on the permeate side, vaporize and pass the membrane, being collected in the nitrogen traps at a constant rate.

The first extraction corresponds to the aromas according to its volume [17,18,23], and the fact is that it presents the lowest refractive index (1.345, which corresponds to an alcoholic degree of 23.4% vol.). Therefore, the content of this first extraction will be added later to the final wine. As can be seen in the Figure 5b, the values for alcohol content are increasing until reaching the fifth extraction, in which the highest value is reached, 35.5% vol. Afterwards the permeate graduation stabilizes, remaining almost constant during the rest of the process. In any case the behavior is very similar to the previous case (Figure 4a,b), but in this case alcohol values in the permeate are sensibly higher than in Case 1 and a longer membrane stabilization time is needed, possibly because the membrane is in contact with wine rather than with an NF permeate that should only contain small molecules and a sensibly lower amount of alcohol.

##### Wine Reconstruction

Once PV is finished, a partially dealcoholized wine (retentate) is obtained. The 2 mL that were obtained from the first PV permeate extraction, which should be the fraction with the highest aromatic content as previously mentioned, were added to this retentate. A total of 2.25 L of treated wine were obtained, which were bottled instantly, avoiding possible oxidations.

#### 3.1.3. Case 3 (D)

In Case, 3, prior to the filtration step, a study was conducted to select the most appropriate membrane for the Dialysis process.

##### Membrane Selection

As commented previously, for this case, several membranes were tested to check their performance in the D process. Same synthetic wine was subjected to D filtration for more than 21 h, with extractions (5 mL) from each side flask (wine and water) every hour to control (by refractometry) the content of alcohol extracted from wine side and which part of it passed to the water side. After 4 h, the process continued all night and a final extraction was made on the filtration end.

Time evolution of the alcohol contents in both wine side and water side flasks are compared in Figure 6 for all membranes studied. Here, it can be clearly seen that Ultracel membrane presents the best results, with a clearly faster alcohol reduction which allows us to obtain the expected reduction of alcohol degree in sensibly lower times.

##### Filtration of Real Wine

According to the results of the previous study, it seems clear that, among the membranes tested, PLGC15005 offers the best performance, at least in terms of alcohol recovery. Therefore, this membrane was selected to perform the D filtration of the actual white wine. For such purpose, and taking into account that OIV does not allows alcohol reductions over 20% of initial value (which means 2% degrees for a wine initially around 10% vol.), the filtration time was reduced to approximately 8 h.

To get enough filtered wine for the sensory testing, D was performed in three sessions with an initial amount of 1.4 L of the original white wine, filtered in each session. All sessions lasted for 8 h, and the resulting wines of the 3 sessions were homogenized and bottled again for further chemical analysis and sensorial testing

### 3.2. Chemical Analysis

The reconstructed wine and the original one were analyzed for the parameters described in Materials and Methods Section and for all 3 cases of study. Corresponding results are presented in Table 2 for all Cases.

Firstly, a consideration must be made about the analysis of the original wines. Since they correspond to different harvests and years, they are expected to have slightly different compositions, reflecting the atmospheric and maturation history of each vintage.

Referring to the change of the parameters from original to filtered wines we can note several aspects as follows. pH remains almost unchanged for all cases, with a slight increase in Cases 1 and 2 and slight decrease in Case 3. Otherwise, total acidity decreased significantly for Cases 1 and 3, and remained constant for Case 2. For Case 1, this decrease may be due to selectivity of the NF membrane to tartaric acid (majority within a wine), as was also observed in previous works [46]. This selectivity to tartaric acid could also be the explanation of the slight reduction of total acidity for Case 3. Normally a reduction in total acidity is accompanied by a certain increase of pH, which did not happen for Case 3. However, it should be considered that pH and total acidity refer to different contributions. While pH accounts for the proton concentration, total acidity is expressed in terms of tartaric acid, one of the acid components of wine, along with malic, citric, and lactic. Then, different behavior of pH and total acidity can be experimentally found. On the other side, volatile acidity increased slightly for Case 1, with a very low decrease in Case 2 and a bit higher for Case 2. Regarding the volatile acidity, a significant increase in the dealcoholized wine can be seen; this may be due to the excessive recirculation (especially during NF step), that caused a high aeration, and to the high temperature at which the nanofiltration retention was maintained during pervaporation. Previous studies [46], showing an increase in the content of acetaldehyde, acetic acid precursor, were observed in wines that underwent a dealcoholization using membranes. As in the case of total acidity, in studies carried out partially removing the sugars from a must of Verdejo, to later elaborate wine with it [47], the volatile acidity of the resulting wine was equal and even lower when made with NF or UF. As demonstrated in other studies [48], the temperature and lighting to which the wine was subjected during the NF and PV processes could favor the increase of the volatile acidity of our wine that is dealcoholized.

The content of free sulfur dioxide in both samples is low for Cases 1 and 2 (not detected for Case 1) while a bit higher for Case 3. This relatively low values of free sulfides is connected with the time passed (from initial bottling to the present study), which is clearly longer in the Cases 1 or 2 (2014 and 2015 vintage, respectively), and also with the homogenization and sedimentation that caused small losses of free sulfides that could have been in the wine. Regarding the total sulfur dioxide, it fell significantly for Cases 1 and 2, remaining almost constant for Case 3.

The most important chemical parameter here is the alcohol content, whose reduction is the objective of this work. For Case 1, alcohol content reduces from 11.20% in the original wine to the final 9.20% in the reconstructed one, thus achieving nearly the maximum allowed reduction. However, this wine was reconstructed with appropriated additions of NF retentate and PV permeate, so the alcohol reduction was forced. Second Case leaded to a lower alcohol reduction (0.9%), while D achieved a gentle alcohol reduction (1.2%) which could be higher, if necessary, just by increasing slightly the filtration time.

Here are insignificant changes on TPI after filtration in Cases 1 and 2, with significant reduction for Case 3. While color increases sensibly in Case 1 and, to a minor extent, in Case 2, Case 3 led to almost equal values of color, which means that the wine does not suffer from oxidation or browning after processing.

### 3.3. Sensory Analysis

#### 3.3.1. Triangular Test of Differences

For all results of triangular test, hits means that the judge was able to discriminate which of the 3 wines presented was different while wrong answers correspond to a failure. Table 3 presents results of a triangular test for all Cases, with the following number of judges: 26 for Case 1, 27 for Case 2, and 33 for Case 3. For the sake of clarity, the vintage of each case must also be included, and finally the number of hits needed to consider both wines (filtered and original), distinguishable with a *p* < 5% level of significance.

In the first two cases it was found that the tasters successfully distinguished between both wines at a significance level of *p* ≤ 0.05, however in Case 3 the tasters were not able to distinguish between the two wines and, therefore there are no perceptible differences between the samples (*p* ≤ 0.05). As can be seen in last row of Table 3, only 14 consumers were able to distinguish both wines, while 17 were needed to be sure of such distinguishability.

In addition, the confidence interval on the proportion of the population that can distinguish the sample was calculated. In Case 1 the upper confidence limit is 0.99 and the lower one is 0.67; in Case 2 the upper confidence limit is 0.51 and the lower one is 0.04; finally, regarding Case 3 the upper confidence limit is 0.34 and the lower one has a negative value. From the statistical analysis of data, we can conclude that the actual proportion of the population capable of distinguishing the samples is between 99 and 67% in Case 1, between 51 and 4% in Case 2 and, between 0 and 34% in Case 3, at a confidence level of 95%. Therefore, the organoleptic differences in Case 3, if any, must be a little noticeable.

#### 3.3.2. Check-All-That-Apply (CATA)

Now, the data obtained during the CATA test will be presented and commented (Table 4).

Case 1: As can be seen from the previous Table, filtered wine is considered as cleaner than original wine, but this is usual after any filtration procedure. On the other hand, the great majority of the aromatic attributes of the dealcoholized wine, minus the herbaceous aroma, were less significant than in the case of the original wine. Likewise, the overall aromatic intensity of the dealcoholized wine was also valued as being considerably lower than that of the original wine. Also, important tasting parameters as volume in mouth and persistence are registered by judges in much lower extent for filtered wine. Finally, the overall valuation of the wines also is less favorable to the treated wine, so we can conclude that the procedure applied in Case 1 results in a worse wine in aromatic, gustatory, and overall aspects.

Contrary to what we have found in our study for yet developed wines, when the dealcoholization of the wines has been carried out by means of a partial reduction of the sugars of the starting musts [47], the differences have been much less sensorial. So, from both sensory tests we can conclude that, in Case 1, treated wine presents important organoleptic differences with original wine which makes this option not very useful for industrial applications.

Case 2: Regarding the sensory attributes related to the visual phase, the control wine has a greater limpidity (expected after filtration), and a less oxidized appearance than the control wine.

In the olfactory phase, the dealcoholized wine seems to have lost part of its aromatic qualities, but not as strongly as in Case 1. Anyway, this leads to sensible damage of its aromatic intensity. These changes resulted in a lower acceptability of the filtered wine with respect to the original one.

Case 3: According to the Cochran test for results of Case 3, none of the attributes presented significant differences between original and treated wine. However, some comments should be made.

It is a bit surprising that the descriptor clean obtained higher attributions for the original wine that for the treated one, since normally filtration results in cleaner wines. In any case, the differences are so small that they could be neglected. Also, Dialysis treated wine is found as more oxidized than the original wine but both have such small punctuations of rust that these differences can be considered as likely experimental errors.

Referring to olfactory parameters, it seems Dialysis filtration has not resulted in a sensible loss of aromatic aspects; even most parameters present some increase. However, whole D wine is reported as having a less intense aroma.

Regarding gustatory aspects, filtered wine has very similar valuations as the original one, and finally this is also reflected in an exactly equal acceptability. It is worth noting that the wine of this year apparently was considered lower quality even in origin, as it only reached 30% acceptance as compared with almost 60% for the other two Cases. It must be remembered that each case corresponded to a different vintage and different winemaking, which reflects the sensitivity of the wine to the initial quality of the grapes.

#### 3.3.3. CATA Comparison of All Cases

For a better discussion of the influence of both membrane-based approaches on final oenological parameters, a comparison of the changes induced for both treatments in the resulting wines is presented in Figure 7. In this figure, the difference between the original and the treated wines is plotted for each parameter. There, a positive answer for a given parameters means that (in the opinion of the judge) the tested sample possesses such a descriptor; the number of positive answers were recorded for each descriptor and for both original and treated wines in the two cases in the study. Then, we can define the change observed in each descriptor according to the following expression:Ch_i_ (%) = 100 × (Tr_i_ − Or_i_)/Or_i_(1)

Here Ch_i_ is the change observed for the i-th descriptor (from those presented in Table 4), Tr_i_ the value recorded for such descriptor (number of hits) for the treated wine, and Or_i_ the similar value obtained for original wine.

These percentage values are plotted in Figure 7 for each descriptor and for the three cases studied.

As can be seen, there are positive and negative differences indicating descriptors which appear more clearly in the treated wine (positive values) while others do not appear or appear with less frequency (negative values).

Clearly for treatments applied in Cases 1 and 2, the resulting wines were associated to the cleanness descriptor. Surprisingly, Case 3 led to an almost negligible change in limpidity, which could indicate this treatment is less aggressive to treated wines.

On the other hand, descriptors associated to a well-made and rounded wine as: volume in mouth, intensity of aroma or persistence, suffered important changes that clearly can be associated to a loss of aroma compounds during the filtration process. Anyway, this loss is lower in the case of the dialyzed wine (which in fact increases some aromatic perceptions) being these differences clearly higher for the case of pervaporated wine and clearly much higher when NF followed by PV was applied. It is clear that the loss of wine descriptors is mostly associated with the pumping needed during filtration, much more intense in the NF stage, while PV, which is a low-pressure process, needs less power of pumping, and finally D, where there is no transmembrane pressure, needs even less aggressive pumping.

Finally, the overall acceptability of the resulting wines is clearly related with the process suffered. In that sense, NF resulted in a strong fall of acceptability (clearly related with sensible losses in most tasting parameters), while PV also reduced acceptability of the wine but in a sensibly lower extent. Finally D treated wine showed same acceptability of the original wine.

## 4. Conclusions

Concerning the three procedures tested for the partial dealcoholization of white Verdejo wines, we can conclude the following:

NF process resulted the most aggressive treatment for the wine, due to the need of pumping (at relatively high pressures) resulting in sensible losses of its organoleptic qualities, still unrecovered after the PV application. This resulted in a reconstructed wine presenting notable sensorial differences with original wine, clearly reflected in the triangular test, where only 3 judges from 26 were unable to distinguish filtered and original wines.

The use of only PV, clearly less aggressive in terms of pumping, lead to a resulting wine with reasonability good chemical comparison with original wine along with sensory analyses better than NF. Certain loss of aroma, persistence, and volume in mouth, of filtered wine, made it less acceptable than the original. Triangular test confirmed this fact, with enough consumers able to distinguish both wines.

Finally, the use of Dialysis for the partial dealcoholization of the wine resulted in reasonably low chemical differences (except alcohol reduction). However, the most interesting feature is that for most sensorial characters tested, filtered wine presented similar marks as original one. On the other side, triangular test allows us to assure both wines are almost indistinguishable from organoleptic point of view.

As a final conclusion, we can state that, in the experimental conditions of our study, and applied to white wine of the Verdejo variety, the Membrane Dialysis process presents notable advantages over PV or combined NF + PV. This clearly better organoleptic behavior could envisage the utility of such process at industrial scale for partial dealcoholization of this kind of white wines, with scarce loss of sensorial and organoleptic properties of the wine and assuring the acceptability of them for potential consumers.

It is important to recall on a point that has so far been overlooked. Once demonstrated the feasibility of Dialysis for alcohol reduction of Verdejo white it is clear that application of this process only makes sense in the case of wines with natural alcohol contents considered excessive for the type of wine or the Appellation of Origin (AO) concerned, which was not the particular case for our wines. In general, the OIV does not allow for the name of wine to be given to products containing less than 8.5% alcohol (which was not the case in any of the treatments here studied). However, the different AOs can enforce more restrictive conditions to their wines. In particular, the Rueda appellation (which covers the different types of white wines made from the Verdejo grape in the region of Castilla y León) only grants this appellation to wines containing a minimum of 11%, so that none of the examples included here could be marketed under this appellation. It is therefore up to the oenologist of each winery to decide whether the starting wine has sufficient alcohol content to merit the treatment discussed here.

Finally, the influence of the alcohol reduction on the stability of the resulting wines needed to be tested in a longer study. These studies would ensure the availability of wines treated by dialysis for subsequent maturation in barrels and the consequent increase in their sensory characteristics and the resulting added value.

## Figures and Tables

**Figure 1 membranes-12-00468-f001:**
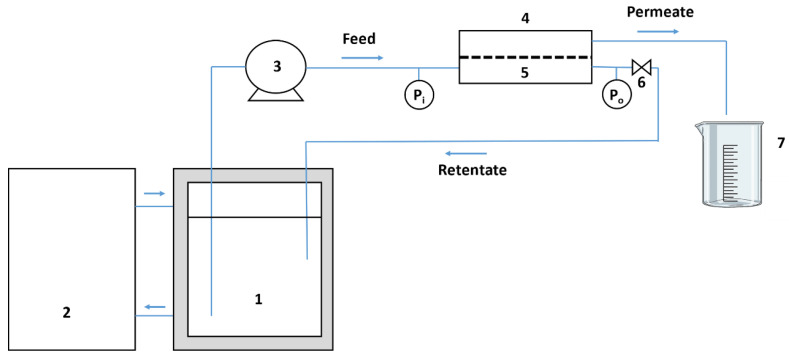
Nanofiltration (NF) experimental system.

**Figure 2 membranes-12-00468-f002:**
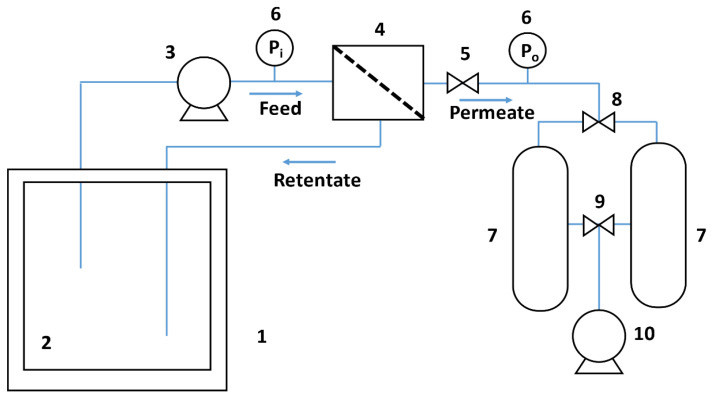
Pervaporation (PV) experimental system.

**Figure 3 membranes-12-00468-f003:**
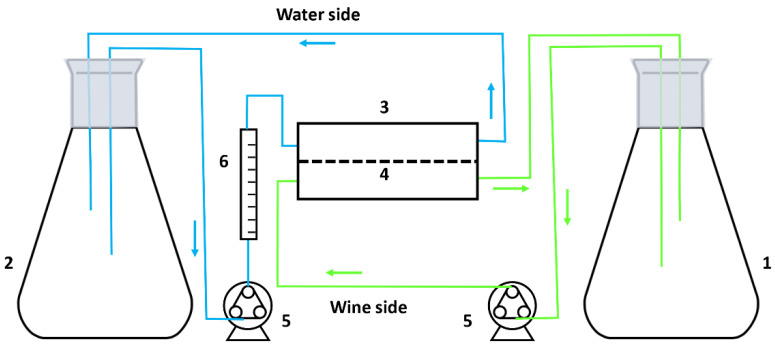
Dialysis (D) experimental system.

**Figure 4 membranes-12-00468-f004:**
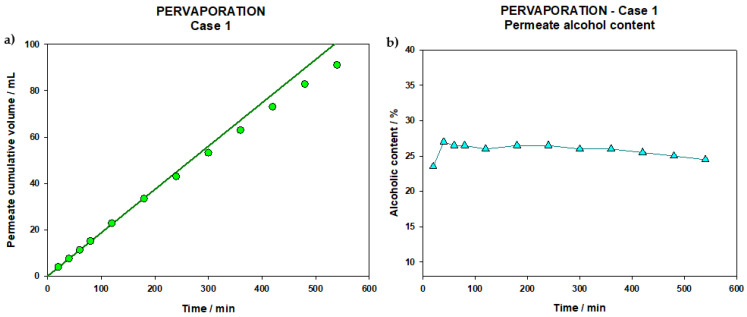
(**a**) Cumulative volume of permeate collected during PV step of Case 1. (**b**) Alcohol content of PV permeate (Case 1).

**Figure 5 membranes-12-00468-f005:**
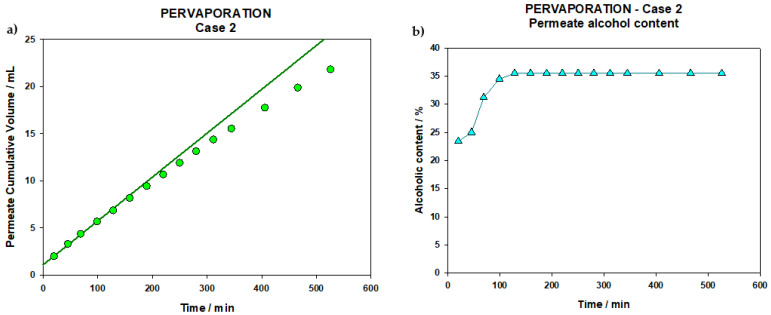
(**a**) Cumulative volume of permeate collected during PV (Case 2). (**b**) Alcohol content of PV permeate (Case 2).

**Figure 6 membranes-12-00468-f006:**
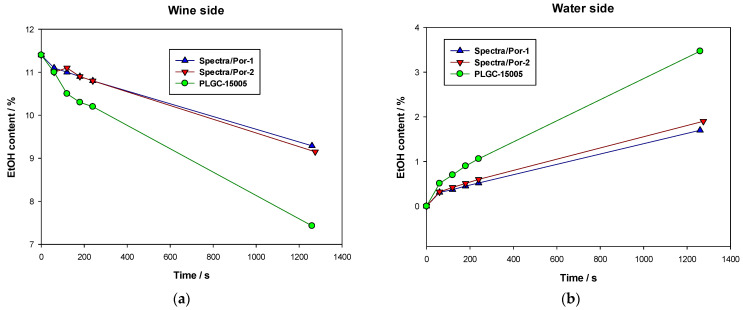
Alcohol content evolution during D for each membrane tested: (**a**) Wine side, (**b**) Water side.

**Figure 7 membranes-12-00468-f007:**
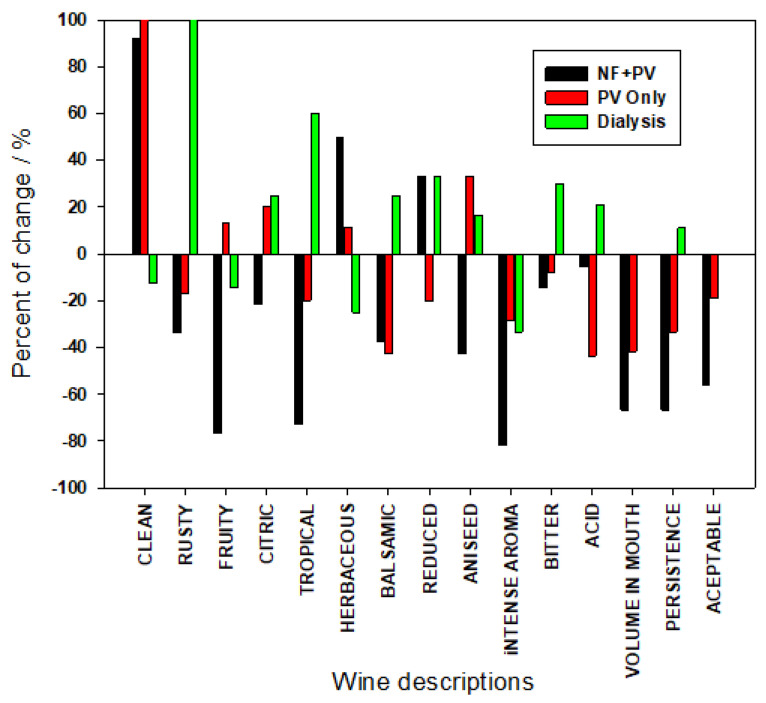
Differences obtained for all parameters tested in the CATA test (in percentage) with the original wines, for all cases studied.

**Table 1 membranes-12-00468-t001:** Nominal characteristics of membranes used.

Process	Membrane	Manufacturer	pH Range	Max. Temperature /(°C)	Max. Pressure /(bar)	MWCO /(kDa)
**NF**	DK-28D	General Electric	3–9	80	30	0.15–3
**PV**	PDMS	Pervatech	1–12	70	5	-
**D**	PLG15005	Merck	2–12	121	5	10
**D**	SpectraPor1	Fisher Sci.	2–12	121	-	6–8
**D**	SpectraPor2	Fisher Sci.	2–12	121	-	12–14

**Table 2 membranes-12-00468-t002:** Mean values and standard deviations of wines chemical composition (for all 3 cases).

	Case 1 (2014 Vintage)		Case 2 (2015 Vintage)		Case 3 (2018 Vintage)	
	Original Wine	NF + PV	Original Wine	PV	Original Wine	D
**pH**	3.17 ± 0.03 ^a^	3.22 ± 0.01 ^a^	2.90 ± 0.03 ^a^	3.00 ± 0.01 ^a^	3.24 ± 0.02 ^a^	3.04 ± 0.01 ^b^
**Total acidity (g/L tartaric acid)**	4.90 ± 0.01 ^a^	4.02 ± 0.02 ^b^	4.40 ± 0.01 ^a^	4.40 ± 0.01 ^a^	4.6 ± 0.06 ^a^	4.2 ± 0.00 ^b^
**Volatile acidity (g/L acetic acid)**	0.16 ± 0.01 ^b^	0.22 ± 0.01 ^a^	0.16 ± 0.01 ^a^	0.13 ± 0.01 ^b^	0.34 ± 0.01 ^a^	0.30 ± 0.02 ^a^
**Color (AU)**	0.103 ± 0.02 ^a^	0.122 ± 0.01 ^a^	0.097 ± 0.02 ^b^	0.105 ± 0.01 ^a^	0.094 ± 0.002 ^a^	0.084 ± 0.003 ^a^
**Alcohol content (% *v*/*v*)**	11.20 ± 0.10 ^a^	9.20 ± 0.10 ^b^	11.50 ± 0.10 ^a^	10.60 ± 0.20 ^b^	9.90 ± 0.00 ^a^	8.70 ± 0.00 ^b^
**Free sulfur dioxide (mg/L)**	nd	nd	3 ± 0.0 ^a^	3 ± 0.0 ^a^	43 ± 2 ^a^	29 ± 6 ^a^
**Total sulfur dioxide (mg/L)**	75 ± 0.02 ^a^	66 ± 0.06 ^b^	109 ± 0.0 ^a^	89 ± 0.06 ^b^	98 ± 2.1 ^a^	103 ± 4.0 ^a^
**TPI (Total Polyphenol Index)**	4 ± 0.1 ^a^	4 ± 0.1 ^a^	4 ± 0.0 ^a^	4 ± 0.0 ^a^	5 ± 0.05 ^a^	4 ± 0.16 ^b^

nd: not detected. ^a,b^: different letters indicate significant differences among samples for each case (*p* < 0.05, Tukey test).

**Table 3 membranes-12-00468-t003:** Data obtained in the triangular test (for all Cases).

	Case 1 (26 Judges)	Case 2 (27 Judges)	Case 3 (33 Judges)
**Hits**	23	14	14
**Failures**	3	13	19
**Vintage**	2014	2015	2018
**Significant number of answers**	14	14	17

**Table 4 membranes-12-00468-t004:** Frequency of citations obtained in the CATA test used by consumers to describe each sample for all cases.

	CASE 1 (2014 Vintage)			CASE 2 (2015 Vintage)			CASE 3 (2018 Vintage)		
		Samples			Samples			Samples	
Number	Terms	Original Wine	NF + PV	Terms	Original Wine	PV	Terms	Original Wine	D
**1**	**Clean ****	12	23	**Clean ****	10	20	**Clean ^ns^**	24	21
**2**	**Rusty ^ns^**	3	2	**Rusty ^ns^**	6	5	**Rusty ^ns^**	1	3
**3**	**Fruity *****	17	4	**Fruity ^ns^**	15	17	**Fruity ^ns^**	14	12
**4**	**Citric ^ns^**	14	11	**Citric ^ns^**	10	12	**Citric ^ns^**	16	20
**5**	**Tropical ****	11	3	**Tropical ^ns^**	5	4	**Tropical ^ns^**	5	8
**6**	**Herbaceous ^ns^**	6	9	**Herbaceous ^ns^**	9	10	**Herbaceous ^ns^**	12	9
**7**	**Balsamic ^ns^**	8	5	**Balsamic ^ns^**	7	4	**Balsamic ^ns^**	4	4
**8**	**Reduced ^ns^**	3	4	**Reduced ^ns^**	5	4	**Reduced ^ns^**	3	5
**9**	**Aniseed ^ns^**	7	4	**Aniseed ^ns^**	6	8	**Aniseed ^ns^**	6	7
**10**	**Aroma intensity ****	11	2	**Aroma intensity ***	14	10	**Aroma intensity ^ns^**	15	10
**11**	**Bitter ^ns^**	14	12	**Bitter ^ns^**	13	12	**Bitter ^ns^**	10	13
**12**	**Acid ^ns^**	18	17	**Acid ****	16	9	**Acid ^ns^**	19	23
**13**	**Mouth volume ****	12	4	**Mouth volume ***	12	7	**Mouth volume ^ns^**	5	5
**14**	**Persistence ****	15	5	**Persistence ***	15	10	**Persistence ^ns^**	9	10
**15**	**Acceptable ****	16	7	**Acceptable ^ns^**	16	13	**Acceptable ^ns^**	10	10

***: indicates significant differences at *p* < 0.001. **: indicates significant differences at *p* < 0.01. *: indicates significant differences at *p* < 0.05. ^ns^: indicates no significant differences (*p* > 0.05) according to Cochran’s Q test.

## Data Availability

The data presented in this study are available on request from the corresponding author.

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
