# Peer review of "Membrane Dialysis for Partial Dealcoholization of White Wines"

_membranes, 2022, doi:10.3390/membranes12050468_

Round 1

Reviewer 1 Report

The paper is well written, the subject is interesting and actual. There are few observations I would like to make:

  1. Please carefully check the spelling within the entire manuscript. Use capital letters only where necessary (e.g. Nanofiltration, Pervaporation, Dialysis, etc.)
  2. Do not use lists for pointing out specific details. Different paragraph is enough for the reader, especially in the Conclusion section. Instead of using numbered lists, I suggest better to number within one single paragraph different ideas/processes used. 
  3. No discussion was addressed regarding the conformity of alcohol reduction to existing regulation. How do authors believe it might affect the wines classifications? At the present moment, we call ``wine`` the fermented beverage made exclusively from grapes which must posses alcohol content higher than 8.5 % v/v. Also, it is forbidden to ``reconstruct`` the alcohol content afterwards (after fermentation). How can industrials handle this? Even it is an extremely interesting topic, researchers must present also the disadvantages or actual limits of the procedure and possibly propose solutions to it. Obviously, solutions must be legal ones. 
  4. What about the lower wines contents impact on wines stability? Refer to this aspect in the Conclusion section.

I recommend minor revision of the paper.

Author Response

Referee # 1

The paper is well written, the subject is interesting and actual. There are few observations I would like to make:

We deeply acknowledge the comments and suggestions from the referee which we have tried to incorporate in the revised version.

  1. Please carefully check the spelling within the entire manuscript. Use capital letters only where necessary (e.g. Nanofiltration, Pervaporation, Dialysis, etc.)

We have revised the spelling, using capital letters only for well-known membrane processes, wine variety, certain procedures and commercial names

  1. Do not use lists for pointing out specific details. Different paragraph is enough for the reader, especially in the Conclusion section. Instead of using numbered lists, I suggest better to number within one single paragraph different ideas/processes used. 

Conclusions have been changed according to your suggestion.

  1. No discussion was addressed regarding the conformity of alcohol reduction to existing regulation. How do authors believe it might affect the wines classifications? At the present moment, we call ``wine`` the fermented beverage made exclusively from grapes which must posses alcohol content higher than 8.5 % v/v. Also, it is forbidden to ``reconstruct`` the alcohol content afterwards (after fermentation). How can industrials handle this? Even it is an extremely interesting topic, researchers must present also the disadvantages or actual limits of the procedure and possibly propose solutions to it. Obviously, solutions must be legal ones. 

According to OIV [7], several methods can be used for the correction of the alcohol content in wines after fermentation. These allowed methods include: Partial vacuum evaporation, Membrane techniques and Distillation. And, in any case, the process can be applied if following conditions are assured:

- This process must not be used on wines with any other organoleptic defects.

- The elimination of alcohol in wine must not be done in conjunction with a modification in the sugar content in the corresponding musts.

- The alcohol content may be reduced by a maximum of 20%.

- The minimum alcoholic strength by volume must comply with the definition of wine.

- The process shall be placed under the responsibility of an oenologist or specialised technician.

As you have correctly pointed out, the most complicated point could be an excessive reduction which makes resulting wine not possible to marketed with such name. In the actual Spanish legislation, this minimum alcohol content is 8.5 % which is achieved in all three cases. Nevertheless more famous Verdejo wines are covered by the AO (Appellation of Origin) Rueda which includes more strict regulations to the wines to be marketed under this AO. In fact, Rueda wines cannot have in any case, an alcohol content lower than 11 %, which certainly create a problem for the industry in the 3 cases here described. Nevertheless, the idea of this work is mostly demonstrate the feasibility of the membrane processes, and specially of dialysis, to accomplish the necessary reduction if, as a consequence of the climate change, the wines obtained in the normal fermentation process, results too alcoholic (which was not the case in any of the vintages here considered). To make clearer this point and following your suggestion, a paragraph has been added in the conclusions section (lines 636-655).

  1. What about the lower wines contents impact on wines stability? Refer to this aspect in the Conclusion section.

A paragraph has been added in Conclusions (lines 658-662).

Reviewer 2 Report

In the manuscript  ‘Membrane Dialysis for partial dealcoholization of white wines’ authors investigate the potential of membrane dialysis as a new technique for the dealcoholization of white wines and compare it to pervaporation and nanofiltration combined with pervaporation, which are used for the same scope. The manuscript is thoroughly and concisely written and could be accepted for publication in Membranes journal after the revision based on the following comments:

Line 30: Use some other term instead of ‘very frequent decreases in rainfall’.

Lines 32-44: Do not use bullets, but incorporate this text in the paragraph.

Lines 51-57: Do not use numbering in this way, but incorporate this text in the paragraph.

Line 76: What do you mean by the word ‘feed’?

Lines 111-132: Do not use numbering in this way, but incorporate this text in the paragraph.

Line 127: The abbreviation D was not used before so here it should be written the full text. Also, I suggest to use the abbreviation MD (for membrane dialysis) as it will be easier to follow through the text.

Lines 199-206: Do not use numbering in this way, but incorporate this text in the paragraph.

Lines 210-214: Do not use numbering in this way, but incorporate this text in the paragraph.

Lines 220 and 223: Do not use bullets, but incorporate this text in the paragraph.

Line 232: Do not start the sentence with pH (because of the lowercase letter).

Lines 251-273: Why has not be used the same wine for all the three treatments?

Lines 260-271: Do not use bullets, but incorporate this text in the paragraph.

Improve the discussion of the results, with more references included in the text.

Line 483: Start the sentence with uppercase H.

Rewrite the conclusions to make the text shorter, more concise, and do not use the bullets.

Author Response

See attached doc

Reviewer 3 Report

General comments: the study compares three membranes approaches – (i) Nanofiltration with Pervaporation, (ii) single step Pervaporation and (iii) Dialysis – for the partial removal of alcohol from experimental Verdejo white wines from three vintages: 2014, 2015 and 2018. Regretfully the three trials were applied to wines from different vintages therefore the comparison among membranes is jeopardised. Although the topic is of actual interest, especially related to the climate change impact in enology, the manuscript needs the following amendments before suitable for publication:

Abstract: rewrite part the abstract including the most significant results detailed in metrics, i.e. numbers.

Line 37-38: rewrite as “Resulting musts have lower acidity, thus an excessive pH, and higher sugar contents with variation in the concentration of polyphenols”.

Line 39-40: provide evidence that consumer reject higher alcohol wines due to the restrictive legislation on alcohol rates allowed when driving.

Line 46-47: the issue of loss in colour apply to red wines, which is not this case, therefore rephrase the sentence.

Line 112: replace the term ‘gentle’ with a more scientific measurable concept.

Line 186 and 202: use superscript character for 10-3 m2 and MgSO4 (and throughout the entire text).

Line 228 and 274: in which year the chemical and sensory analysis were done, respectively.

Line 242: given the several compounds that affect the refractive index measurement in wine (e.g. sugars, acids, glycerol, EtOH, etc.) provides the equations used to account for the potential interferences.

Line 274: use a single consistent terminology throughout all text for defining the person involved in the sensory evaluation choosing among consumers, semi-trained or trained assessors.

Figure 6: add the time unit.

Table 2: remove the row ‘samples’; use consistent symbol for decimals (i.e. point, not comma); define the measurement unit of ‘color’; report alcohol % as v/v; for total sulfur dioxide report the method reproducibility; fully explain why, in case 3, after D treatment the total acidity drops whereas pH increases;

Table 3: add a row with the significant number of assessors required to give a significant answer at 5% level, i.e. 14, 14 and 17.

Table 2, 3 and 4: add the year of vintage, i.e. 2014, 2015, 2018 for each wine.

Author Response

See attached doc

Round 2

Reviewer 2 Report

The manuscript has been substantially improved, but the Conclusion section should be considerably shortened to make it more focused and concise.

Reviewer 3 Report

On materials and methods section it must be indicated the period in which the analysis of wines have been done (e.g. 2019?)

Line 430-431: a more thorough explanation is needed about the changes in total acidity and pH of the sample 3: this is critical to accept the manuscript for publication.

Round 3

Reviewer 3 Report

wine chemistry is a complex topic including wine acidity (which further understanding requires advanced knowledge)